WG-Storm: a resource-aware scheduler for distributed stream processing engines

Ali Rizwan 1
Muhammad Asif masifqadri2000@yahoo.com 1
Aleem Muhammad 2
Shafiq Omair 3
1 Department of Software Engineering, National University of Computer and Emerging Sciences, Islamabad , Islamabad , Punjab , Pakistan
2 School of Science, Coventry University , Coventry , United Kingdom
3 Carleton University , Ottawa , Ontario , Canada
Somani Arun
Electronic publication date: 2025 Apr 25
Publication date: 2025
Volume: 11
Electronic Location ID: e2767
Received 2024 Sep 19; Accepted 2025 Feb 25
Copyright: ©2025 Ali et al.
Copyright year: 2025
Copyright holder: Ali et al.
License: This is an open access article distributed under the terms of the Creative Commons Attribution License, which permits unrestricted use, distribution, reproduction and adaptation in any medium and for any purpose provided that it is properly attributed. For attribution, the original author(s), title, publication source (PeerJ Computer Science) and either DOI or URL of the article must be cited.
License URL: https://creativecommons.org/licenses/by/4.0/

Keywords: Resource aware, Heterogeneous cluster, Stream processing engine, Distributed computing, Optimized scheduling

Funding: The authors received no funding for this work.

==============================
Stream processing engines (SPEs) allow applications to process a large amount of data in real-time. However, to schedule big data applications; the SPEs create several challenges regarding resource utilisation, dynamic configurations, heterogeneous environment, resource awareness, load balancing, etc. As the volume of data increases over time, it also poses a challenge to predict the resource and application requirements for processing. All these factors play an important role, they can cause problems in achieving maximum throughput due to inefficiency in any of them. Most SPEs ignore the topology’s structure, which may minimise throughput during scheduling and may increase network latency. In this article, a topology-aware and resource-aware scheduler (named WG-Storm) is proposed based on a directed acyclic graph (DAG) that enhances resource usage and overall throughput using efficient task assignment. WG-Storm is built on Apache Storm. Results are generated using the two linear topologies and compared with the five state-of-art schedulers including A3-Storm, Default, Isolation, Multi-tenant, and Resource-aware. The experimental results show up to 30% increased throughput using the least required computing resources in a heterogeneous cluster.

Introduction

In recent years, real-time processing has gained attention. Stream processing allows us to process data in real-time as it is being processed quickly after receiving it. This allows the user to input the data into the analysis tool immediately after generating the data and obtain instant analysis results (see Fig. 1). Examples of existing frameworks include Apache Kafka (Hiraman, Viresh & Abhijeet, 2018), Apache Flink (Akil, Zhou & Röhm, 2017), Apache Storm (Iqbal & Soomro, 2015), Apache Samza (Zhuang et al., 2016), etc. For stream processing, Apache Storm is a popular framework (Iqbal & Soomro, 2015) due to its wide adoption, maturity, and community support (Hickey, 2010; Qi & Rodriguez, 2021). Apache Storm’s default scheduler is an even scheduler, which distributes tasks to nodes in a round-robin fashion without analyzing the communication between nodes, which introduces a performance bottleneck in terms of load balancing and proper utilization of resources. It is static scheduling means when a job is submitted to a system, the default scheduler creates a static scheduling plan. As a result, the scheduler does not detect changes in the data stream for a distributed environment. In addition, it does not consider memory and CPU resources that may cause system instability. To achieve high throughput of real-time processing with appropriate resource utilization and low latency, we need to find an optimal task arrangement and adjust the requirements accordingly.

Figure 1 Overview of real-time stream processing architecture (Layka et al., 2013).

The default scheduler uses a round-robin technique to assign tasks evenly among nodes. Figure 2 shows a job that is scheduled using a round-robin algorithm. The round-robin scheduler will assign T2 tasks on machine N1 which is a CPU task, even though another powerful machine is present in the cluster. Similarly, other tasks are also assigned in the same fashion which might cause network latency and inefficient resource utilization which resulted in poor performance due to the ignorance of inter-node communication (Zhuang et al., 2016). This motivates the design of a scheduler that minimizes unnecessary network costs by balancing workloads.

Figure 2 Tasks assignment using default Storm scheduler.

Different researchers (Muhammad & Aleem, 2021; Muhammad, Aleem & Islam, 2021; Liu et al., 2019) proposed schedulers that increase the throughput and utilization of resources. However, they (Muhammad, Aleem & Islam, 2021; Zhou et al., 2020; Liu et al., 2019; Souravlas & Anastasiadou, 2020) consider only CPU bound jobs. Some researchers (Souravlas & Anastasiadou, 2020; Bilal & Canini, 2017; Liu & Buyya, 2017) proposed schedulers by considering the homogeneous environment that may lead to cost-efficiency and resource wastage. These aspects play an important role in gaining performance. Therefore, ignoring any aspect can lead to poor performance. So, there is a need to propose a scheme that considers all possible aspects to improve system throughput and proper resource utilization. In this work, a WG-Storm scheduler has been proposed for heterogeneous environments to address these issues. The primary contributions of this work are:

• A thorough analysis of the current state-of-the-art to identify the limitations of existing schedulers.

• Development of a novel scheduling algorithm for heterogeneous computing environments.

• Introduction of WG-Storm, which utilizes the topology structure in a resource-aware manner to improve throughput and resource utilization.

• Conduct experiments and evaluations to compare its performance with state-of-the-art scheduling heuristics.

The remainder of the article is arranged as follows. ‘Related work’ presents the research work done in this domain. ‘Proposed methodology: WG-Storm’ explains the proposed framework. ‘Experiments’ outlines the experimental evaluation. The results are provided in ‘Results comparison and discussion’. Finally, the last section concludes the article with possible future directions. Portions of this text were previously published as part of a preprint (https://doi.org/10.21203/rs.3.rs-2985470/v1).

Related Work

To evaluate the effectiveness of existing schedulers, we identified several key aspects that influence the performance of stream processing engines. These aspects include topology awareness, which ensures that task dependencies and communication patterns are considered; resource awareness, which accounts for the availability and capacity of hardware resources; and heterogeneous environment support, which handles varying computational capabilities across nodes. Additionally, features such as I/O bounds and dynamic scheduling are critical for addressing real-time data processing and system adaptability. These aspects serve as the basis for analyzing prior research in this domain, as discussed in the next section.

In recent years, different researchers have proposed their scheduler to enhance the throughput of Apache Storm. Liu & Buyya (2017) proposed D-Storm which solves the bin-packing problem by using a greedy algorithm. This scheduler monitors topologies at run-time which reduces inter-node communication. Scheduler rescheduled topologies whenever resource contention is detected. However, the authors do not consider the network topology and scheduler also implemented on homogeneous clusters. D-Storm also has the cold-start issue (some traffic is required initially). Similarly, as a result, it is slower than Default Scheduler to make a scheduling plan.

T3-Scheduler is presented by Eskandari et al. (2018) based on topological structure and a communication-aware two-level scheduler that can find most communicating tasks and then assign these tasks at the same node for properly utilizing nodes. T3-Scheduler, at the first level, divides the topology into various parts according to the communication frequency of the task. Therefore, it reduces the communication between nodes by putting together highly communicating tasks. At the second level, it finds the most suitable allocation by putting highly communicative tasks in the same worker process to overcome the communication within the worker processes in the node. The scheduler also solves the heterogeneous problem, but for linear and star types, the average throughput is slightly improved.

Al-Sinayyid & Zhu (2018) proposed the maximum tuple processing rate (MTPR) algorithm. Through the use of dynamic methods, the Storm topology jobs are efficiently scheduled based on computing requirements, data transmission requirements, and the capacity of the underlying cluster resources. Along the critical path obtained from the job. The approach increases the system throughput and reduces the time incurred on the bottleneck. They increase the processing speed of tuples, but they do not have a different DAG structure in a larger Storm cluster.

Liu et al. (2019) proposed an adaptive online scheme that determines the number of resources efficiently needed for each instance without any resource wastage. To minimize the number of affected workers, the authors proposed the resource cost-aware algorithm. Their approach also enforces the resource allocation decisions made by their scheduler. They also see the negative impact of co-locating stateless executors and stateful executors. This scheduling scheme is plugged into the storm and in experiments that achieve better performance than other solutions. However, they ignore the memory and I/O bounds jobs.

Mortazavi-Dehkordi & Zamanifar (2019) proposed a framework, that examines the topology to determine the output size of its operators for analysis, and partitions. The operators assigned to thread computing units and threads that are identified are assigned to processes. The authors also proposed a scheduler for dynamic environments (offline schedulers). The offline scheduler assigns processes to the available nodes. The purpose of a scheduler is to reduce the latency and balance the operator workload. However, the authors did not consider memory in resources.

In 2020, Al-Sinayyid & Zhu (2020) proposed a scheduler named MT-scheduler which maximizes the throughput of the system by using dynamic programming. MT-scheduler uses a dynamic programming technique that maps a DAG onto the heterogeneous systems. The results of this technique show increased system throughput of the system compared with the Round Robin (RR)-Algorithm. By utilizing the dynamic programming methods; the authors proposed a polynomial-time heuristic solution. The authors realized the performance optimization of the system by decreasing the computing and transfer time.

Nasiri et al. (2020) proposed a heterogeneity-aware scheduling algorithm that takes into account the different computing capabilities of heterogeneous machines. The proposed algorithm finds a proper number of vertices and maps them to the most suitable node. By increasing the topology rate they scale up the application graph over the cluster. Their algorithm maximizes the throughput and ensures the over-utilization of the node. The scheduler can recalculate the number of instances at run time whenever any state changes in the cluster. They also proposed the CPU prediction model that predicts the CPU utilization of tasks.

Zhou et al. (2020) presented a scheduling technique, called TOSS, which deals with tightly bound components. By using a tuning mechanism, the scheduler also balances the workload in the deployment stage. TOSS ensures the reduction of rescheduling overhead by decreasing rescheduling events and inter-node traffic. The scheduler can speed up the stream data process by reducing rescheduling overhead. This can identify potential edges with a lot of communication on a static topology. TOSS groups most communication edges and then allocates these groups in the same slot directed by self-tuning parameters.

Souravlas & Anastasiadou (2020) proposed a method based on pipeline modular arithmetic (PMOD scheduler), which is based on each node using tuples at the same time to process only from another node. The scheduler organizes all necessary operations in a pipelined manner, such as tuple packing, tuple transmission, and tuple processing. As a result, the overall execution time is decreased compared to other approaches. The scheduler maximizes load balancing but also increases throughput and minimizes buffer requirements.

In cloud computing, resource provisioning and scheduling face challenges due to resource heterogeneity, dispersion, and energy consumption concerns. This article (Kumar et al., 2020) introduces a modified binary particle swarm optimization (BPSO) algorithm, to efficiently allocate applications among virtual machines while optimizing quality of service (QoS) parameters like makespan time, energy consumption, and execution cost. BPSO enhances exploration and exploitation capabilities, addressing BPSO’s limitations. Through Pareto theory, it balances conflicting parameters, yielding non-dominated solutions. Performance analysis against baseline algorithms using Cloudsim demonstrates BPSO’s efficiency in optimizing influential parameters under user budget constraints. However, the comparison with synthetic datasets limits generalizability to real-world scenarios. Further research is needed to explore additional QoS parameters like availability, reliability, and service level agreement (SLA) violation.

In 2021, Muhammad & Aleem (2021) used a greedy algorithm for task assignment. The proposed method maps the topology according to the computing requirements and the computing power of the machines. All unassigned tasks are arranged with respect to their communication traffic that requires high computing power. These executors are assigned to nodes with respect to their computational power which can ensure enhanced throughput and resource utilization.

Eskandari et al. (2021) proposed a heuristic scheduling algorithm that increases throughput in both heterogeneous and Homogeneous clusters. Their approach effectively finds most communication tasks by using graph partitioning algorithms and allowing the use of mathematical optimization software to find effective task assignments. In the case where optimization software cannot be used, backup heuristics are also provided. It partitions the application graph attractively according to the capacity of heterogeneous nodes and then allocates the partitions according to the capacity of the node.

Muhammad, Aleem & Islam (2021) proposed the TOP-Storm, which optimizes the resource usage of Storm in heterogeneous clusters. Existing job schedulers have commonly tried to minimize inter-node network communication and resource usage. TOP-Storm maps the topology according to the computational requirements of the topology. Unassigned executors are arranged with respect to their communication traffic. Logical mapping involves task grouping which is done using DAG and physical mapping is done by assigning a highly communicating executor’s group to a node based on the node’s computation power.

Qi & Rodriguez (2021) introduced a dynamic scheduling strategy based on graph partitioning and real-time traffic data monitoring. The scheduler monitors the exchange rate of a tuple within different communication tasks. Assign tasks according to the resource requirements and the traffic pattern to achieve high throughput and low latency. The proposed approach is a two-level technique. First, the topology graph is assigned a weight based on the type of communication. At the second level, the schedulers find multiple partitions and also find unfeasible partitions, by these partitions the authors achieve load balancing of the tasks and minimize tuple communication. After identifying the best partition, the scheduler assigns the tasks in the cluster.

Sun et al. (2021) proposed a model to determine the cumulative processing delay and computing power of each operator and also the similarity of bolts. The authors also considered bandwidth and the latency of communication between inter-workers. The author’s approach also reduces communication latency by putting highly communicating tasks into the same worker and increases overall throughput. However, the authors ignore the I/O bounds and memory bounds jobs.

Cloud computing is widely used for its diverse services, with virtual machine (VM) configuration being a key focus for researchers. The challenge lies in optimizing VM placement in dynamic environments, impacting QoS parameters. An optimistic approach is proposed (Sharma, Kumar & Samriya, 2022) to enhance QoS and reduce configuration overhead by consolidating VMs and employing task migration. Experiments in CloudSim show significant improvement over other methods like PSOCOGENT, Min-Min, and FCFS. Strengths include improved QoS, reduced overhead, and efficient scheduling, while weaknesses include reliance on synthetic workload data and limited consideration of other QoS factors like reliability and availability.

In 2022, MF-Storm (Muhammad & Qadir, 2022) is presented which is based on a famous max-flow min-cut algorithm to achieve maximized throughput. MF-Storm considers communication demand as well as available resources while scheduling in two steps. First, the application’s DAG is partitioned to minimize inter-partition communication. Then, these partitions are mapped to nodes according to the computing power of the nodes.

Kumar et al. (2023) presents a secure, self-adaptive resource allocation framework utilising an enhanced Spider Monkey Optimization (SMO) algorithm to meet dynamic workload demands in cloud computing. The proposed framework integrates advanced encryption for infrastructure security. It employs the enhanced SMO algorithm for optimal resource allocation, improving parameters such as time, cost, load balancing, energy consumption, and task rejection ratio. Experimental results on CloudSim show superiority over particle swarm optimization (PSO), gravitational search algorithm (GSA), artificial bee colony (ABC), and IMMLB. The main contribution lies in ensuring infrastructure security while dynamically allocating resources. However, it may overlook factors like reliability and fault tolerance, and the evaluation is limited to simulation environments.

In 2023, a deep Q-learning-based scheduling mechanism was introduced (Mangalampalli et al., 2023) to address challenges in cloud task scheduling. It outperforms baseline algorithms like FCFS and EDF in reducing makespan, SLA violations, and energy consumption, as demonstrated through CloudSim simulations. Strengths include effective optimization and the use of machine learning, while weaknesses involve reliance on synthetic datasets and limited consideration of QoS parameters beyond makespan and energy consumption.

Provisioning computer resources over the internet has popularized the cloud computing paradigm. However, challenges like efficient resource allocation and avoiding under or over-provisioning are crucial, known as cloud computing load forecasting. Recurrent neural network (RNN) models for cloud forecasting are underexplored. This study (Bacanin et al., 2023) uses long short-term memory (LSTM) models, with and without attention layers, for cloud load time-series forecasting. Key contributions include proposing LSTM and LSTM with attention layers for cloud-load forecasting and addressing the underexplored RNN-based models. Introducing a modified PSO algorithm for hyperparameter tuning, showing improvements over original and contemporary optimizers. The proposed VMD-LSTM-MPSO technique shows promising results, with a mean squared error of 0.00159 and R2 of 0.77624. This method enhances resource allocation planning, cost optimization, scalability, proactive maintenance, and fault detection for cloud providers. Limitations include not testing complex networks, and exploring only a subset of metaheuristics.

Recent advancements in networking technologies have shifted focus towards distributed cloud-based services, necessitating effective management of computation resources to maintain cost efficiency and service quality. This research (Predić et al., 2024) addresses the gap in utilizing recurrent neural networks (RNNs) with attention mechanisms for cloud computing by proposing a methodology for forecasting cloud resource load. It Utilizes RNNs with attention mechanisms and decomposition techniques (VMD) for cloud-load forecasting. Introducing a modified PSO metaheuristic for hyperparameter tuning, demonstrating statistically significant improvements over original and contemporary optimizers. The proposed method shows great potential in accurately forecasting cloud load, with optimized models outperforming competing approaches and providing valuable insights for decision-making in cloud service management. Limitations include computational demands and the exclusion of other RNN variants like LSTM, bidirectional long short-term memory (BiLSTM), and gated recurrent unit (GRU).

Table 1 presents the summary of the related work. The Storm scheduler considers all available worker nodes for scheduling tasks in a cluster. This leads to higher computing costs and low resource utilization. It evenly distributes the workload to the work processes of the entire cluster without considering the communication between nodes and processes, which may cause delay and lower throughput (Aniello, Baldoni & Querzoni, 2013). Several researchers have introduced scheduling approaches to enhance the throughput of the Apache Storm. The majority of articles (Ficco, Pietrantuono & Russo, 2018; Muhammad, Aleem & Islam, 2021; Zhou et al., 2020; Liu et al., 2019; Souravlas & Anastasiadou, 2020; Mortazavi-Dehkordi & Zamanifar, 2019), improved resource utilization and throughput but considered only CPU Bound Jobs. Most researchers ignore the Memory bound and I/O bounds jobs. Some researchers (Ficco, Pietrantuono & Russo, 2018; Eskandari et al., 2021; Muhammad, Aleem & Islam, 2021; Li, Zhang & Luo, 2017) also ignore bandwidth impact that will affect the system throughput. Some authors did not consider the heterogeneous environment (Liu & Buyya, 2017; Zhou et al., 2020), which may lead to resource wastage. Hence, there is a need to design a scheduler that should be resource-aware and topology-aware by considering the different aspects (i.e., data transfer rate, heterogeneous environment, Memory, and I/O bounds jobs).

Table 1 Literature review summary.

Scheduling aspects	Muhammad & Qadir (2022)	Zhuang et al. (2016)	Muhammad & Aleem (2021)	Muhammad, Aleem & Islam, 2021	Liu et al. (2019)	Souravlas & Anastasiadou (2020)	Zhou et al. (2020)	Sun et al. (2021)	
Topology-aware	Yes	Yes	Yes	Yes	Yes	Yes	Yes	Yes	
Resource-aware	Yes	Yes	Yes	Yes	Yes	Yes	Yes	Yes	
Heterogeneous	Yes	Yes	Yes	Yes	No	Yes	Yes	Yes	
Dynamic	Yes	Yes	Yes	Yes	Yes	Yes	Yes	Yes	
Traffic-aware	Yes	Yes	Yes	No	Yes	Yes	Yes	Yes	
CPU Bounds	Yes	Yes	Yes	Yes	Yes	Yes	No	Yes	
Memory Bounds	No	No	No	No	No	No	Yes	Yes	
I/O Bounds	No	No	No	No	No	No	No	No	
Bandwidth	No	No	No	No	No	No	Yes	Yes	

Figure 3 WG-Storm system architecture.

Figure 4 WG-Storm proposed scheduler.

Proposed methodology: WG-Storm

The WG-Storm scheduler allocates jobs on a heterogeneous cluster to increase throughput and improve resource utilization. It assigns topologies by considering the computation power of the available nodes and the computational requirements of the topologies. Figure 3 represented the system architecture of the WG-Storm. WG-Storm realizes topology awareness and efficient resource scheduling by including new features in the Storm framework. It consists of four modules (1) Monitoring, (2) Scheduler, (3) Network Agent, and (4) Resource Manager. WG-Storm runs on the master node, which receives input from another component and makes key decisions related to topology mapping. The network agent runs on each node and provides the resource manager with information about the hardware configuration (RAM, number of cores, frequency, etc.) used for computing power and traffic between tasks. The resource manager computes the computing power of each node. By considering the computing power and computing requirements of the topology the WG-Storm maps the topology on available nodes. Most powerful machines are used first, to achieve higher throughput. The architecture of the WG-Storm scheduler consists of six main steps (as shown in Fig. 4).

• Monitoring: The WG-Storm monitors the connectivity of the tasks (topology awareness), the data transfer rate between tasks (communication), and resource availability (such as computing power, bus speed, memory, and physical location of nodes) to ensure proper use of resources;

• Constructing a weighted graph: At this level, a weighted graph is constructed by using monitoring steps to provide the WG-Storm scheduler with a global view of tasks and communication load;

• Pairs: At this level, by applying the BFS algorithm on the weighted graph we observe that the task group should co-locate within nodes by finding a subgraph of most of the communicating tasks;

• Logical grouping: Topology’s DAG is used for the task’s assignment. In most cases, the communication tasks are placed as close as possible;

• Sorting: At this level sort available nodes according to the computation power (most powerful at the start) and also sort groups of frequent communication tasks;

• Physical grouping: The group is assigned to the slot. The slots are allocated starting from the node with the most computing power.

After monitoring all the task’s connectivity and data transfer rates between tasks, the WG-Storm scheduler constructs a weighted graph based on the data transfer rate as shown in Fig. 5. First, the WG-Storm scheduler builds the vertices of every component of the topology. The scheduler then creates an edge between the vertices that are connected and assigns the weight to them based on the collected information. In Fig. 5, the T1, T2, T3, T4, T5, and T6 represent the tasks of topology and the edge weight represents the data transfer rate between two tasks. We traverse the graph using the BFS algorithm. The BFS algorithm traverses a graph one level at a time. This search algorithm is used for finding the shortest path between the vertex of the graph. By using this algorithm, we create a pair of tasks and then logically map these pairs. Logical mapping involves task assignments to slots by using the pairs. It continuously maps the tasks to slots until all slots are mapped.

Figure 5 The construction of weighted graph (Qi & Rodriguez, 2021).

________________________________________________________________________________________________ Algorithm 1 WG-Storm   1:  graph ← CreateGraph(executors)      ⊳ Create a graph based on available      executors   2:  components ← BFSTraversal(root)        ⊳ Perform Breadth-First Search      traversal starting from the root   3:  taskPairs ← createTaskPairs(components)  ⊳ Create task pairs from the      identified components   4:  sortedNodes ← GetandSortNodes()          ⊳ Get and sort nodes based on      certain criteria   5:  tasksGroups ← logicalGrouping(taskPairs) ⊳ Perform logical grouping of      task pairs   6:  while all tasks in taskGroups are not assigned do  7:       SCHEDULE(taskGroups,sortedNodes)    ⊳ Map groups to slots and      schedule   8:  end while   _______________________________________________________________________________________________

WG-Storm algorithm

WG-Storm consists of three main steps: graph traversal, executor assignment, and slot assignment, designed to optimize resource utilization and maximize throughput. The detailed process is outlined in Algorithm 1 and is closely tied to the performance improvements demonstrated in the results.

• Graph traversal: WG-Storm begins by creating a weighted graph of unassigned executors using the createGraph procedure ( Algorithm 1 , Line 1). The graph captures task connectivity and communication patterns, where the edge weights represent the data transfer rate between tasks. This step minimizes inter-node communication costs by providing a global view of task dependencies. The traversal process employs the BFS algorithm to identify task pairs that communicate most frequently. For example, in Fig. 6, BFS starts at a root task (e.g., T1), exploring its immediate neighbors (T2 and T4) in Level 1, followed by deeper connectivity layers such as T3, T5, and T6 in subsequent levels. By focusing on high-communication links like (T1, T2) and (T2, T3), WG-Storm ensures that such tasks are co-located whenever possible, thereby reducing communication delays and improving throughput.

• Executor assignment: The next step is to assign executors to computational nodes based on their capacity. WG-Storm retrieves cluster information ( Algorithm 1 , Line 4) and computes the computation power of each node as a product of its cores, frequency, and RAM ( Algorithm 2 ). Nodes are sorted in descending order of their computational power, ensuring that high-demand tasks are assigned to the most capable nodes.

• Slot assignment: Finally, WG-Storm performs slot assignment by mapping task pairs (created in the graph traversal step) to physical slots. Task groups are assigned starting from the most powerful node, continuing until all slots are filled ( Algorithm 1 , Lines 6–8). This systematic approach ensures balanced load distribution while keeping communication-intensive tasks co-located, thereby reducing network latency. By integrating real-time task connectivity and resource availability into the scheduling process, WG-Storm achieves consistent performance gains. Its combination of dynamic graph traversal, node sorting by computational power, and efficient slot assignment ensures reduced inter-node traffic and maximized throughput.

Once task pairs are created in the graph traversal step, WG-Storm groups these tasks according to the computational power of the nodes present in the cluster. Algorithm 2 dynamically calculates and ranks the computational power of each node to prioritize task assignments to the most capable nodes. This process directly contributes to the improved throughput and resource utilization observed in the experimental results.

Figure 6 The BFS traversal.

____________________________________________________________________________________________ Algorithm 2 Sort Nodes   1:  nodes ← getClusterNodes()          ⊳ Get all available nodes in the cluster   2:  while all n in Nodes do              ⊳ Iterate through each node in the list   3:       cores ← n.getCores()     ⊳ Retrieve the number of cores for the current      node   4:       frequency ← n.getFrequency() ⊳ Retrieve the frequency of the current      node   5:       RAM ← n.getRAM()       ⊳ Retrieve the RAM capacity of the current      node   6:       computationPower ← cores × frequency × RAM           ⊳ Calculate      computation power for the current node   7:  end while  8:  Nodes ← Sort()    ⊳ Sort nodes in descending order based on computation      power   9:  return Nodes                               ⊳ Return the sorted list of nodes ____________________________________________________________________________________________

First, the algorithm gathers hardware specifications of all nodes in the cluster, including the number of cores, processor frequency, and RAM ( Algorithm 2 , Lines 3–5). The computation power of each node is calculated as the product of these parameters, representing the node’s capacity to handle tasks efficiently. For example, nodes with higher core counts, faster frequencies, and larger RAM are ranked higher. Sorting the nodes in descending order of computational power ensures that tasks demanding higher resources are assigned to the most powerful nodes first. This approach minimizes under-utilization of high-performance nodes, leading to better load balancing and reduced latency.

After sorting, WG-Storm maps task groups to the available slots on each node ( Algorithm 1 , Lines 6–8). By considering both task requirements and node capabilities, the algorithm ensures that the maximum number of tasks can be efficiently scheduled. By dynamically ranking and utilizing nodes based on their hardware specifications, Algorithm 2 enables WG-Storm to achieve significant performance gains in both throughput and resource utilization, as evidenced across multiple configurations in the experimental evaluation.

WG-Storm introduces a significant advancement in task scheduling compared to existing schedulers. Unlike the Default Scheduler, which statically assigns tasks using a round-robin approach, leading to inefficient resource utilization and increased inter-node traffic, WG-Storm dynamically assigns tasks based on resource availability and inter-task communication patterns. In contrast to the Multitenant Scheduler, which allocates resources per user rather than per topology, WG-Storm ensures optimal resource utilization by prioritizing highly communicative task pairs and assigning them to the most powerful nodes. Similarly, the Isolation Scheduler, while effective in dedicating resources to specific topologies, often underutilizes available resources and lacks mechanisms for inter-task communication optimization, which WG-Storm achieves through its weighted graph and BFS-based task pairing. Compared to the Resource-Aware Scheduler, WG-Storm not only considers resource availability but also integrates node computation power (calculated as a combination of cores, frequency, and RAM) and dynamically adjusts task assignments based on real-time workloads. Finally, while A3-Storm Scheduler focuses on reducing inter-executor traffic in heterogeneous clusters, it lacks WG-Storm’s real-time adaptability and comprehensive resource awareness, as WG-Storm continuously monitors task connectivity and dynamically updates task groupings to maximize throughput and minimize inter-node traffic.

Experiments

Experimental setup

To evaluate the performance of the WG-Storm, a heterogeneous cluster is configured with one ZooKeeper, one Nimbus, and five supervisors. On each node Ubuntu 18.04.4 LTS 64-bit,) is installed with network connectivity. Every node has a Java OpenJDK 11(11.0.11 2021-04-20) and uses Storm 2.3.0 (Apache Storm, 2024) and Zookeeper 3.7. Table 2 presents experimental environment hardware configurations including the RAM, Processor, Disk Space, FLOPS/sec, IP address, etc for all nodes.

Table 2 Hardware configurations of the employed experimental cluster.

Hostname	IP address	Processor configuration	Disk space (GBs)	FLOPs/cycle	
Nimbus-Server	172.17.37.165	Intel(R) Core(TM) i5-4210 CPU @ 3.300 GHz × 8	500	6,593	
Node-A	172.17.46.166	Intel(R) Core(TM) i5-4210 CPU @ 3.300 GHz × 8	1,000	6,813	
Node-B	172.17.45.236	Intel(R) Core(TM) i5-4210 CPU @ 3.300 GHz × 8	500	6,813	
Node-C	172.17.39.63	Intel(R) Core(TM) i5-4210 CPU @ 3.300 GHz × 8	500	6,813	
Node-D	172.17.58.35	Intel(R) Core(TM) i5-10700 CPU @ 2.90 GHz × 8	1,000	6,813	
Node-F	172.17.44.69	Intel(R) Core(TM) i7-10510U (8) @ 4.900 GHz × 8	500	6,813	

State-of-the-art schedulers

WG-Storm is compared with four default storm schedulers (Apache Storm, 2024) and one state-of-art scheduler:

1. The Default Scheduler uses a round-robin algorithm to distribute a topology workload equally into worker processes without considering any priority, which leads to less throughput and higher response time.

2. In Isolation Scheduler you can specify which topologies will be “isolated” so they can run on dedicated machines in a cluster.

3. In Resource-aware Scheduler when scheduling the topology the Apache Storm considers resource availability as well as resource requirement of workloads.

4. Multitenant Scheduler provides resource allotment per user instead of per topology.

5. A3-Storm Scheduler (Muhammad & Aleem, 2021) minimize the resource usage for the heterogeneous cluster. It schedules topology using the computing power of node and inter-executor traffic.

Topology selection

The Storm has various types of topology structures for example, linear topology layout (Fig. 7), diamond topology layout (Fig. 8), and star topology layout (Fig. 9). The assessment of WG-Storm’s performance focuses on two linear topologies that are capable of efficiently processing data and making informed decisions. Specifically, we employed the topologies shipped with the Apache Storm suite (Apache, 2024), widely recognized and employed by researchers as benchmarks (Xie et al., 2017) are the following:

Figure 7 Layout of linear topology.

Figure 8 Layout of diamond topology.

Figure 9 Layout of star topology.

1. Word Count Topology (Apache Storm Community, 2022)

2. Exclamation Topology (Apache Storm Community, 2022)

The following metrics have been used for performance evaluation:

1. Throughput that represents the number of tuples processed

2. Resources used represents the number of nodes used for topology execution

It is clarified that improvements in throughput inherently reflect reduced latency, as these metrics are inversely related. In the context of real-time processing systems like WG-Storm, increasing throughput means the system can process more tasks in a given time frame, which reduces the waiting and processing time for each task. Consequently, this leads to lower end-to-end latency. By focusing on throughput in our evaluation, we implicitly demonstrate WG-Storm’s capability to reduce latency, thus ensuring its applicability for latency-sensitive applications such as financial trading, IoT, and real-time analytics.

Table 3 represents the different configurations for experiments, In configuration 01, the required numbers of slots for scheduling both typologies word count and exclamation are three, and available slots per supervisor are three, we have a total of five supervisor nodes, a total (three slots per supervisor ×five nodes) is 15 slots. Similarly, for configuration 02 and configuration 03, the required slots are three. In configuration 04, 05, and 06, the required number of slots is four and the available slots per supervisor node are three, four and five, respectively. The total slots available are 15, 20, and 25 for configurations 04, 05, and 06. In configuration 07, the available slots per node are three and the required slots for executing topology are five, for configuration 08 and configuration 09 the required number of slots is five and, the available is four for all state-of-art schedulers. In configuration 09 for scheduling topology, the required slots are five and the total available is 25. Based on these scenarios, we perform all the experiments with five state-of-art schedulers.

Table 3 Different configuration for experiment.

Number of slots required by topology	Number of available slots per supervisor node	
	3	4	5	
3	Configuration 01	Configuration 02	Configuration 03	
4	Configuration 04	Configuration 05	Configuration 06	
5	Configuration 07	Configuration 08	Configuration 09	

Results Comparison and Discussion

This section represents the evaluation of the WG-Storm with five different schedulers. In this evaluation, the required and available slots varied from three to five for each topology to investigate the effect on throughput. For example, in configuration 05 we have four slots per supervisor and we have a total of five nodes in a cluster, e.g., 20 (four slots ×five nodes) for scheduling the topology. Each experiment is executed for 15 min and reading is noted after 30 seconds.

Comparison with the other schedulers

The comparison is performed with four default storm schedulers using configurations 1–9 (in Table 3). Performance comparison using the word count topology with all configurations and with exclamation topology respectively as shown below. In configuration 01, we have five supervisor nodes, three slots per node (total 15) and for executing word count topology three slots are required. Figure 10 shows the traffic (inter-executors) produced using configuration 01. Here proposed WG-Storm improves 11% on average throughput with respect to default scheduler and improves 15% as compared with the Multi-tenant scheduler and almost 50% performs better than the isolation scheduler.

Figure 10 Performance comparison—word count topology (configuration 01).

In Fig. 11, the performance comparison between WG-Storm, four default storm schedulers and A3-Storm scheduler using configuration 02. Configuration 02 has five supervisor nodes, three slots per supervisor (total 15) for executing word count topology four slots are required. Average throughput improved by 10% and 20% for default and Multi-tenant schedulers, respectively and WG-Storm outperformed all other schedulers in configuration 02.

Figure 11 Performance comparison—word count topology (configuration 02).

The comparison between WG-Storm and four schedulers including A3-Storm executed using configuration 03. In configuration 03, in which we required five slots for executing topology out of 15 slots. WG-Storm improved average throughput by 4% as compared to A3-Storm and 10% for the default Apache storm scheduler (see Fig. 12).

Figure 12 Performance comparison word count topology (configuration 03).

Figure 13 illustrates the comparisons between the WG-Storm scheduler with A3-Storm and four default storm schedulers. WG-Storm scheduler performed almost equally with A3-Storm and improved average throughput by approximately 17% for the default Apache storm scheduler using configuration 04. In configuration we use five supervisors nodes, each node has four slots (total four ×five) are 20, but the required slots for executing the word count topology are three. Similarly in Fig. 14, we compared the WG-Storm scheduler with others using different configurations. WG-Storm scheduler performs 15% better than A3-Storm and 8% for the default scheduler by using configuration 05. For executing the topology, four slots are required out of 20 (5 supervisors node, four slots per node).

Figure 15 shows the performance comparisons for configuration 06, in which we have five supervisor nodes, each node contains four slots total of 20 slots available for executing the topology, and the required slots are five for executing topology. Similarly, Fig. 16 shows a comparison using configuration 07, in which average throughput increased up to 10% with the A3-storm scheduler and 5% for the default Apache storm scheduler. Similarly, for configuration 08 (Fig. 17) and 09 (Fig. 18), the overall throughput is better than other schemes.

Figure 13 Performance comparison—word count topology (configuration 04).

Figure 14 Performance comparison—word count topology (configuration 05).

Figure 15 Performance comparison—word count topology (configuration 06).

Figure 16 Performance comparison word count topology (configuration 07).

Figure 17 Performance comparison—word count topology (configuration 08).

Figure 18 Performance comparison—word count topology (configuration 09).

Comparison of results with other schedulers by executing the exclamation topology shown in Figs. 19–27. We compare WG-Storm with other scheduling algorithms with all configurations (1–9, Table 3). In configuration 01, WG-Storm improves 3% on average throughput concerning the A3-Storm and improves 50% as compared with the default Apache storm scheduler. Similarly in configurations 02 and 03 results are shown in Figs. 20 and 21 respectively.

Figure 19 Performance comparison—exclamation topology (configuration 01).

Figure 20 Performance comparison—exclamation topology (configuration 02).

Figure 21 Performance comparison—exclamation topology (configuration 03).

Figure 22 shows the average throughput achieved by executing the exclamation topology in which we get 19% better throughput compared to the default scheduler. WG-Storm achieved equal throughput as compared with the A3-Storm using configuration 04. Similarly, Figs. 23–27 show the performance comparison between WG-Storm and five state-of-the-art schedulers by executing the exclamation topology. The overall comparison with all configurations using word count and exclamation topologies is shown in Figs. 28, and Fig. 29, respectively.

Figure 22 Performance comparison—exclamation topology (configuration 04).

Figure 23 Performance comparison—exclamation topology (configuration 05).

Result analysis

WG-Storm performs better in terms of resource usage and throughput as compared to the other schedulers. If we ignore the resource availability and communication pattern the results that we achieve will be difficult. The default scheduler decreases throughput and increases inter-node traffic because it uses the round-robin approach and engages the maximum nodes. WG-Storm uses the available slots of one supervisor and then moves to the next one. So, that way WG-Storm achieved 25% for exclamation topologies and 30% for word count. The topologies in the Isolation scheduler execute on the dedicated cluster. It generates different mapping and focuses on improving resource utilization instead of throughput which is why it is not performing well in all configurations (see Table 3) in terms of throughput for both topologies.

Figure 24 Performance comparison—exclamation topology (configuration 06).

Figure 25 Performance comparison—exclamation topology (configuration 07).

Figure 26 Performance comparison—exclamation Topology (configuration 08).

Figure 27 Performance comparison—exclamation topology (configuration 09).

Figure 28 Average throughput achieved for wordcount topology using different configurations.

Figure 29 Average throughput achieved for exclamation topology using different configurations.

The overall results for word count topology are shown in Fig. 28. The WG-Storm placed most communicating tasks closer and considered the inter-node traffic causing higher throughput. Other schedulers cannot consider the computational requirements while capturing the resources. Figure 29 shows the results for exclamation topology under different configurations. The overall performance improved as compared with other schedulers in all configurations (1–9).

Conclusions and Future Work

In this article, WG-Storm is presented to improve the average throughput based on inter-executors traffic. WG-Storm considered the DAG’s topology, nodes’ computing power, and communication between tasks for slots mapping. WG-Storm increased throughput and minimized inter-node communication by getting information about nodes including their physical presence. WG-Storm is compared with five state-of-art schedulers including (A3-Storm) and showed improvement in throughput up to 25% to 30%, respectively. We considered two linear topologies for scheduling (i.e., count and exclamation topology). In the future, we will consider the I/O bound jobs, in this work we consider CPU and memory-based jobs. Similarly, we will try to perform all experiments on the cloud (i.e., AWS) as well. Other typologies like diamond and star can be used to perform the experiments. Furthermore, this work can be extended for other SPEs as well such as Apache Spark, Apache Flink, etc.

Supplemental Information

Supplemental Information 1 WGStorm Scheduler

Supplemental Information 2 Raw Data

Additional Information and Declarations

Competing Interests

Author Contributions

Data Availability

The authors declare there are no competing interests.

Rizwan Ali conceived and designed the experiments, performed the experiments, performed the computation work, prepared figures and/or tables, and approved the final draft.

Asif Muhammad analyzed the data, prepared figures and/or tables, and approved the final draft.

Muhammad Aleem conceived and designed the experiments, authored or reviewed drafts of the article, and approved the final draft.

Omair Shafiq analyzed the data, authored or reviewed drafts of the article, and approved the final draft.

The following information was supplied regarding data availability:

The code and raw data are available in the Supplemental Files.

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
