# Peer review of "WG-Storm: a resource-aware scheduler for distributed stream processing engines"

_PeerJ Computer Science, doi:10.7717/peerj-cs.2767_

## Round 0.1 · original submission · Major Revisions

Please note the concerns expressed by the reviewers on the research work, experimentation methodology, and results are extensive. They require full attention. I would expect that you update the research work, experiments, and then report the finding. Please also include a separate document detailing your response to specific comments made by the reviewers along with the changes you made in the paper. The revised paper will be sent for a second review.

If you choose not to undertake the revision, please let us know. Thanks for your interest in the journal.

**Language Note:** The review process has identified that the English language must be improved. PeerJ can provide language editing services - please contact us at [email protected] for pricing (be sure to provide your manuscript number and title). Alternatively, you should make your own arrangements to improve the language quality and provide details in your response letter. – PeerJ Staff

Reviewer 1 ·

Basic reporting

Technical quality of the submission is good in both English and stylistic and formal arrangement. The graphs included in the submission should be in vector format (not as bitmap images) to allow better printing. The literature review is sufficient although its findings are not adequately utilized in the text to support presented ideas.

I appreciate comparison of the related work in Tab. 1 "Literature review summary". However, the aspects described in the table (especially those missing features in the most of schedulers, such as "I/O Bounds") should be introduced before the literature review and evaluation/comparison methodology should be established. Also there should be more detailed description of how the missing features from the comparison will be addressed in the proposed approach (before or after its description).

The algorithm presented in the Sec. 3 is described clearly, however, it is not so clear how it differ from approaches taken by the previous works. It should go much further than at a trivial calculation of "computationPower" (dynamic loads, a prediction, operating with RT features of Storm flows, etc.).

Experimental design

While Sec. 5 shows a lot of impressing graphs of efficiency of the proposed scheduling algorithm, those are not supported by data published in the paper (although there could be in the included attachments... but they are missing also there). I would expect better experimental setup (beyond basic serial and parallel flows) and statistical numbers proving correct approach with established statistical analysis methods. Without those, the graphs are just mere illustrations, not scientific results. The included Java code of the WGStorm class implementing IScheduler interface in org.apache.storm.scheduler package seems ok, however, it does not prove the presented results and there are not data to make the experiment and scientific results reproducible.

Validity of the findings

Without more detailed experimental results, possible impact and novelty of the proposed approach cannot be assessed. The description of the algorithm proposed in the submission does not prove the presented results.

Cite this review as

Reviewer 2 ·

Basic reporting

## Summary

This article introduces WG-Storm, a resource-aware and topology-aware scheduler for Apache Storm, a Stream Processing Engine (SPE). The scheduler enhances resource utilization and throughput by efficiently assigning tasks using a Directed Acyclic Graph (DAG). WG-Storm is compared to five other schedulers and demonstrates up to a 30% increase in throughput while minimizing resource usage.

## Language

There are issues with phrasing. Certain sentences are unnecessarily length and could introduce ambiguity. For example, in the sentence: “This is a motivation for designing a scheduler that minimizes unnecessary network costs by balancing workloads.” The phrase “This is a motivation for designing” could be more naturally phrased as “This motivates the design of” to improve clarity and flow.

Some parts of the text are somewhat repetitive. For instance: “In this work, a WG-Storm scheduler has been proposed for heterogeneous environments to address these issues.” Such phrases appear multiple times in the text.

## Literature Review

The article presents a long list of related work and provides a brief summary of such work. I appreciate the good survey of related work. On the other hand, the article does not contextualize such work in relation to the work presented. I found it hard to understand what the key contributions made by the authors were, when placed next to other work. I think the authors should expand on how they position their work relative to the vast amount of related work. I think this change is **very important**. Without a clear understand of the contributions, a reader will struggle to appreciate the work.

As one example, there is work by Eskandari that the authors cite but the discussion is insufficient: Eskandari’s schedulers seem more theoretically grounded in mathematical optimization and offer deeper insights into how communication patterns and system heterogeneity affect performance. WG-Storm, while simpler in design, offers a more heuristic-based, practical solution that could scale well in larger, less structured environments. However, like Eskandari’s work, it doesn’t address large-scale failures or highly dynamic cloud-like environments.

## Article Structure

The structure of the article is reasonable. I would, personally, position the work and highlight what is new and leave a detailed discussion of prior work to later. Providing the related work very early becomes a distraction from the core ideas.

## Other Aspects

I would think the article is mostly self-contained. The issue of adding context relative to prior work is the missing piece. There are no formal proofs, but that is not the point of this article.

Experimental design

The experimental work could be enhanced. The authors have studied two simple benchmark applications, which feels limiting given that SPEs have been around for many years now. The authors could also consider other topologies.

Many real-time stream processing systems handle data with varying arrival rates, bursts, and unpredictable patterns, such as those found in IoT systems or financial trading platforms. While this paper’s experiments focus on throughput, they have not addressed the critical issue of latency sensitivity, which is crucial in time-sensitive applications. In many cases, lower latency is as important, if not more, than high throughput. Evaluating the scheduler under highly variable, bursty data streams would give a more complete picture of its practical utility.

Stream processing systems deployed in real-world environments often span hundreds or even thousands of nodes, and they experience failures and node churn. The experiments in this paper are conducted on a relatively small-scale cluster. Scalability testing is crucial to ensure that WG-Storm can maintain its efficiency with increasing cluster sizes and loads. Similarly, fault tolerance was not addressed in the experiments, which is a key requirement in distributed systems where node failures, network issues, and other disruptions occur frequently.

Validity of the findings

The findings appear to be valid given the caveats around the extent of experimentation evaluation. The conclusions are reasonable in that light.

Additional comments

## Add Dynamic and Real-World Workloads

Introduce dynamic workloads where data rates, system load, and node availability change over time. Simulating real-world conditions, such as burst traffic, dynamic resource contention, and node failures, would test the scheduler’s resilience and adaptability.

This would showcase WG-Storm’s ability to handle real-world stream processing challenges like fault tolerance, elasticity, and load balancing in unpredictable environments.

## Evaluate on Larger, More Heterogeneous Clusters

Expand the experiments to larger-scale deployments and more heterogeneous clusters, potentially including cloud environments like AWS or Google Cloud. This would allow WG-Storm to be tested under a more realistic distribution of resources (e.g., a mix of high-performance and low-performance nodes).

Testing on large-scale, real-world cloud platforms would demonstrate the scheduler’s scalability, ability to handle resource heterogeneity, and practicality in deployment.

## Measure Latency in Addition to Throughput

Include latency measurements (e.g., end-to-end latency, task scheduling latency) in the experiments to provide a more comprehensive evaluation of WG-Storm’s performance. Many real-time applications prioritize low latency, and the scheduler’s impact on this metric is crucial. (Look at the mean and 95th percentile, for instance.) This would provide insights into whether WG-Storm is suitable for latency-sensitive applications, such as financial trading, IoT, or real-time analytics, where the timeliness of processing is as important as throughput.

Cite this review as

---

## Round 0.2 · accepted · Accept

Based on the satisfactory update, I am happy to accept the paper. Thanks for your contribution to and interest in the journal.

Reviewer 1 ·

Basic reporting

This is my second review of the submission after its revision so I will make it brief and address only issues that were identified in the first review.

Technical quality is good in the current version. The literature review has been improved as well as its utilization in the another sections of the paper and its is acceptable now.

The description of the algorithm in the Sec. 3.1 was also significantly improved.

Experimental design

A proof for experimental results has been provided as source data for the graphs presented in the paper. Still, better than the measurements would be, e.g., scripts that were used for the performance tests and monitoring. Anyway, the artefacts are sufficient.

Validity of the findings

Valid, no problem here.

Additional comments

The revised version of the submission can be accepted.

Cite this review as